# Using Real-World Data to Determine Health System Costs of Ontario Women Screened for Breast Cancer

Nicole Mittmann [1,2], Soo Jin Seung [3,*], Christina Diong [4], Jodi M. Gatley [4], Michael Wolfson [5], Marie-Hélène Guertin [6], Nora Pashayan [7], Jacques Simard [8] and Anna M. Chiarelli [9,10]

1 Department of Pharmacology & Toxicology, University of Toronto, 1 King's College Circle, Toronto, ON M5S 1A8, Canada
2 Sunnybrook Research Institute, Sunnybrook Health Sciences Centre, Toronto, ON M4G 3M5, Canada
3 HOPE Research Centre, Sunnybrook Research Institute, 2075 Bayview Avenue, Toronto, ON M4N 3M5, Canada
4 ICES Central, 2075 Bayview Avenue, Toronto, ON M4N 3M5, Canada
5 School of Epidemiology and Public Health, University of Ottawa, 600 Peter Morand Crescent, Ottawa, ON K1G 5Z3, Canada
6 Institut National De Santé Publique Du Québec, 945, av Wolfe, Quebec City, QC G1V 5B3, Canada
7 Department of Applied Health Research, University College London, 1-19 Torrington Place, London WC1E 7Hb, UK
8 CHU de Québec-Université Laval Research Center, 2705 Boul Laurier, Quebec City, QC G1V 4G2, Canada
9 Ontario Health, 525 University Avenue, Toronto, ON M5G 2L3, Canada
10 Dalla Lana School of Public Health, University of Toronto, 155 College Street, Toronto, ON M5T 3M7, Canada
* Correspondence: soojin.seung@sunnybrook.ca; Tel.: +1-416-480-4851

**Abstract:** Our study was to determine breast cancer screening costs in Ontario, Canada for screenings conducted through a formal (Ontario Breast Screening Program, OBSP) and informal (non-OBSP) screening program using administrative databases. Included women were 49–74 years of age when receiving screening mammograms between 1 January 2013 to 31 December 2019. Each woman was followed for a screening episode with screening and diagnostic components, and costs were calculated as an average cost per woman per month in 2021 Canadian dollars. The final cohort of 1,546,386 women screened had a mean age of 59.4 ± 7.1 years and ~87% were screened via OBSP. The average total cost per woman per month was $136 ± $103, $134 ± $103 and $155 ± $104 for the entire, OBSP and non-OBSP cohorts, respectively. This was further disaggregated into the average total screening cost per month, which was $103 ± $8, $100 ± $4 and $117 ± $9 per woman, and the average total diagnostic cost per woman per month at $219 ± $166, $228 ± $165 and $178 ± $159. for the entire, OBSP and non-OBSP cohorts, respectively. These results indicate similar screening costs across the different cohorts, but higher diagnostic costs for the OBSP cohort.

**Keywords:** breast cancer screening; costs; mammograms; diagnostics

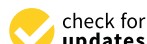

## 1. Introduction

While the Ontario Breast Screening Program (OBSP) screens both high risk and average risk women, the latter group of women are screened with biennial mammography, or screened annually due to certain breast cancer risk factors such as family history and dense breasts [1]. Although the OBSP began in 1990, program data were centralized from 2000 onwards when Cancer Care Ontario (CCO) developed a provincial breast screening database, the Integrated Client Management System (ICMS), to facilitate the operation, monitoring and evaluation of OBSP screening and assessment [2]. Key data elements from the OBSP database are routinely transferred to the Institute for Clinical Evaluative Sciences (ICES), which is an independent, non-profit research institute whose legal status under Ontario's health information privacy law allows it to collect and analyze health care and demographic data, without consent, for health system evaluation and improvement.

Therefore, the OBSP data can be linked with other provincial health services databases to conduct breast screening-related research. For example, Ontario has a population of 14.6 million residents and provides publicly funded health care services through the Ontario Health Insurance Plan (OHIP) [3].

The cost-effectiveness of breast screening programs has been a topic of debate. A few studies [4,5] using simulation models that were modified to reflect the Canadian experience have generally indicated that the more mammograms a woman has during her life, the greater the financial cost to the health care system, but the greater the gain in life-years and quality-adjusted life-years. Recently, the multi-institutional study "Personalized risk assessment for prevention and early detection of breast cancer: Integration and implementation" was initiated, where the objective of one key activity is to determine the real-world health system resources and costs associated with breast cancer screening in Ontario using provincial databases [6]. This provided the study team with the opportunity to link women with average risk for breast cancer who were screened both in a formal provincial screening program like OBSP and non-formal (e.g., non-OBSP) in order to identify the screening and diagnostic cost components as well as the overhead and administrative program costs. We recognize that this is only one part of a standard cost-effectiveness analysis and acknowledge this limitation, with the intent to categorize all the screened women as true negative, true positive, false negative and false positive in our next analysis in order to provide clinical context.

The objective of this study was to determine breast cancer screening costs in Ontario for screenings conducted through the OBSP as well as non-OBSP screenings using administrative databases

## 2. Materials and Methods

### 2.1. Study Design

A longitudinal, population-level study was conducted in women aged 49–74 years of average risk for breast cancer between 1 January 2013 and 31 December 2019, with follow-up data until 31 August 2020, using real-world, population-level data from provincial (Ontario) databases. This study period was selected since the screening mammogram codes became available in October 2010; however, full adoption was achieved in 2013 (A. Chiarelli, personal communication, 21 October 2022).

### 2.2. Study Cohort

Included women had to reside in Ontario, be community-dwelling (i.e., not residing in long-term care), and between 49 and 74 years of age when they received their screening mammograms between 1 January 2013 to 31 December 2019; only the earliest screening mammogram during this period was retained for each woman. Screening mammograms were identified using physician billings from the OHIP database using the OHIP fee code X172 for a unilateral screening mammogram or X178 for a bilateral screening mammogram [7].

Women who were potentially at higher risk for breast cancer such as those who had any screening between the ages of 30–48 years, had a prior breast cancer diagnosis in the Ontario Cancer Registry (OCR) except for ductal carcinoma in situ, or had a mastectomy or breast implants based on OHIP billings (see Table S1 in Supplementary Material) were excluded. Women were censored from any further costing on the date of any of the following events (i.e., cost up until that date): death, 75th birthday, diagnosis for breast cancer (except for ductal carcinoma in situ), mastectomy, breast implants, loss of OHIP eligibility, or admission to a long-term care facility (LTC).

Each woman was followed for a "screening episode", an 8-month period starting on the date of their screening mammogram based on the maximum amount of time anticipated to complete diagnostic procedures following an abnormal mammogram finding (A. Chiarelli, personal communication, 1 April 2021). Women were classified as having had an OBSP screening if they had an OBSP screening date within ±7 days of their screening

mammogram service date in OHIP. If women received a screening mammogram in OHIP but were not found in the OBSP database, they were categorized as having had a non-OBSP screening. Women were further classified as having a positive screening episode if they received any follow-up diagnostic procedure within the 8-month follow-up period including a diagnostic mammogram, ultrasound, computerized tomography (CT) scan, magnetic resonance imaging (MRI), biopsy, and/or genetic consultation for definitions (and costs) of diagnostic procedures (see Table S2 in Supplementary Materials). Women were classified as having a negative screening episode if no diagnostic procedures had occurred. We grouped screening episodes in the following four mutually exclusive categories when reporting costs: OBSP negative (Screening only), OBSP positive (Screening+Diagnostic), non-OBSP negative (Screening only), and non-OBSP positive (Screening+Diagnostic). It should be noted that for the purposes of the study objectives, a 30-day period was used to report the results. A schematic can be viewed in the Supplementary Material.

### 2.3. Data Sources

We utilized several population-based health administrative datasets for Ontario. The Registered Persons Database (RPDB) includes individual health card number, date of birth, sex, postal code, and death date (where applicable). The Postal Code Conversion File (PCCF) database allows linkage to postal codes of residence to determine other census geographic identifiers such as urban/rural flag and neighbourhood income quintile. The Local Health Integration Network (LHIN) database contains postal code lookup tables used to determine patient LHIN of residence in Ontario. The OBSP database includes key data elements of screening encounters. The OHIP database includes physician visits, diagnoses, and fees for health professionals including general practitioners, medical oncologists, radiation oncologists, and other specialists. The OCR database contains all cancer diagnoses. The Discharge Abstract Database (DAD) contains records of inpatient hospitalizations including diagnoses and procedures. The National Ambulatory Care Reporting System (NACRS) database contains records of outpatient clinic visits such as emergency department and cancer care clinics, as well as diagnoses and procedures. The Continuing Care Reporting System (CCRS) LTC database includes clinical and demographic information on residents receiving facility based LTC services. These datasets were linked using unique encoded identifiers and analyzed at ICES. ICES is an independent, non-profit research institute whose legal status under Ontario's health information privacy law allows it to collect and analyze health care and demographic data, without consent, for health system evaluation and improvement.

### 2.4. Statistical and Costing Analysis

Statistical analyses were performed in SAS software, version 9.4 (SAS Institute Inc., Cary, NC, USA). Demographics are summarized by counts and percentages for categorical variables and by mean and standard deviation (SD), median and interquartile range (IQR), and minimum and maximum values for continuous variables. We retrieved demographic information including age, neighborhood income quintile, rurality, and LHIN of residence. Comorbidity was described using the Charlson Comorbidity score based on healthcare utilization in the past two years from DAD and NACRS [8].

An analysis of resource use and direct medical costs was undertaken to understand the current spending associated with receiving screening mammograms in Ontario. Overall total and mean costs associated with screening episodes per women are reported in 2021 Canadian dollars (when possible), using a macro-based costing methodology called GETCOST that is available from ICES. The ICES GETCOST macro was used to determine resource utilization and the total direct medical costs for this cohort. This costing methodology has been described in a previous publication [9]. Costs specific to the administration of screenings the OBSP were estimated based on documentation shared by the OBSP program and personal communications (see Table S3 in Supplementary Materials). Mean costs are

compared between groups using independent samples t-tests using SAS software, and we described costs per women per month.

*2.5. Sensitivity Analysis*

Since screening programs are also available in other provinces, we worked with the Quebec breast screening program coordinators but could not access Quebec-specific breast screening data. Instead, the decision was made to only use Quebec-specific unit costs (see Table S4 in Supplementary Materials) applicable to the screening mammograms, all diagnostic procedures including biopsies and assessments. No other Quebec screening program information (e.g., average/high risk age groups) was used so the same Ontario screening program information was used in the sensitivity analysis.

**3. Results**

*3.1. Demographics*

There were 1,895,312 women between the ages of 49–74 years who received a screening mammogram between 1 January 2013 and 31 December 2019. 348,926 women were excluded for being high risk or for being at a long-term care facility, leaving the final cohort at 1,546,386. Mean age of this cohort at screening was 59.4 years (±7.1 years). Almost 87% of the women were screened in the OBSP Program. The four cohorts included Negative OBSP (74%), Positive OBSP (13%), Negative Non-OBSP (10%), and Positive Non-OBSP (3%). Table 1 shows the baseline characteristics for the cohorts.

**Table 1.** Characteristics of women aged 49–74 in Ontario from 1 January 2013 to 31 December 2019, by screening status.

| Baseline Characteristics | Variable Value | Negative OBSP Screening | Positive OBSP Screening | Negative Non–OBSP Screening | Positive Non–OBSP Screening | Total |
|---|---|---|---|---|---|---|
| Number of eligible women | Sample Size | N = 1,144,442 | N = 195,695 | N = 163,031 | N = 43,218 | N = 1,546,386 |
| Year of index screening | 2013 | 405,948 (35.5%) | 51,722 (26.4%) | 61,382 (37.7%) | 14,085 (32.6%) | 533,137 (34.5%) |
| | 2014 | 316,093 (27.6%) | 42,092 (21.5%) | 43,641 (26.8%) | 8799 (20.4%) | 410,625 (26.6%) |
| | 2015 | 129,525 (11.3%) | 25,569 (13.1%) | 24,154 (14.8%) | 6171 (14.3%) | 185,419 (12.0%) |
| | 2016 | 86,676 (7.6%) | 20,810 (10.6%) | 13,073 (8.0%) | 4638 (10.7%) | 125,197 (8.1%) |
| | 2017 | 77,028 (6.7%) | 19,445 (9.9%) | 8811 (5.4%) | 3592 (8.3%) | 108,876 (7.0%) |
| | 2018 | 67,118 (5.9%) | 18,265 (9.3%) | 6562 (4.0%) | 3211 (7.4%) | 95,156 (6.2%) |
| | 2019 | 62,054 (5.4%) | 17,792 (9.1%) | 5408 (3.3%) | 2722 (6.3%) | 87,976 (5.7%) |
| Age at screening (years) | Mean (SD) | 59.7 (7.0) | 58.0 (7.0) | 59.3 (7.5) | 57.7 (7.6) | 59.4 (7.1) |
| | Median (Q1–Q3) | 59 (53–65) | 56 (52–63) | 58 (52–65) | 56 (51–64) | 59 (53–65) |
| | Min–Max | 49–74 | 49–74 | 49–74 | 49–74 | 49–74 |
| Screen age group (years) | 49–54 | 342,716 (29.9%) | 81,698 (41.7%) | 56,012 (34.4%) | 19,182 (44.4%) | 499,608 (32.3%) |
| | 55–59 | 253,115 (22.1%) | 39,452 (20.2%) | 32,879 (20.2%) | 7850 (18.2%) | 333,296 (21.6%) |
| | 60–64 | 227,587 (19.9%) | 31,825 (16.3%) | 28,132 (17.3%) | 6382 (14.8%) | 293,926 (19.0%) |
| | 65–69 | 198,094 (17.3%) | 27,138 (13.9%) | 25,591 (15.7%) | 5433 (12.6%) | 256,256 (16.6%) |
| | 70–74 | 122,930 (10.7%) | 15,582 (8.0%) | 20,417 (12.5%) | 4371 (10.1%) | 163,300 (10.6%) |
| | Missing Data | 1050 (0.1%) | 222 (0.1%) | 178 (0.1%) | 55 (0.1%) | 1505 (0.1%) |
| Rural | N | 998,812 (87.3%) | 174,970 (89.4%) | 146,545 (89.9%) | 39,798 (92.1%) | 1,360,125 (88.0%) |
| | Y | 144,580 (12.6%) | 20,503 (10.5%) | 16,308 (10.0%) | 3365 (7.8%) | 184,756 (11.9%) |
| | Missing Data | 2534 (0.2%) | 433 (0.2%) | 315 (0.2%) | 107 (0.2%) | 3389 (0.2%) |
| Neighbourhood income quintile | 1 (low) | 193,266 (16.9%) | 34,872 (17.8%) | 28,849 (17.7%) | 8305 (19.2%) | 265,292 (17.2%) |
| | 2 | 220,907 (19.3%) | 38,027 (19.4%) | 32,702 (20.1%) | 8616 (19.9%) | 300,252 (19.4%) |
| | 3 | 230,514 (20.1%) | 39,089 (20.0%) | 32,844 (20.1%) | 8458 (19.6%) | 310,905 (20.1%) |
| | 4 | 239,721 (20.9%) | 40,453 (20.7%) | 34,068 (20.9%) | 8624 (20.0%) | 322,866 (20.9%) |
| | 5 (high) | 257,500 (22.5%) | 42,821 (21.9%) | 34,253 (21.0%) | 9108 (21.1%) | 343,682 (22.2%) |
| Charlson Comorbidity | 0 | 459,506 (40.2%) | 78,935 (40.3%) | 63,739 (39.1%) | 17,562 (40.6%) | 619,742 (40.1%) |
| | 1 | 54,514 (4.8%) | 9073 (4.6%) | 7703 (4.7%) | 1991 (4.6%) | 73,281 (4.7%) |
| | 2+ | 29,438 (2.6%) | 9789 (5.0%) | 4344 (2.7%) | 2383 (5.5%) | 45,954 (3.0%) |
| | No hospitalization | 600,984 (52.5%) | 97,898 (50.0%) | 87,245 (53.5%) | 21,282 (49.2%) | 807,409 (52.2%) |
| Episode follow–up (months) | Mean (SD) | 7.9 (0.6) | 7.7 (1.4) | 7.9 (0.7) | 7.6 (1.5) | 7.9 (0.8) |
| | Median (Q1–Q3) | 8 (8–8) | 8 (8–8) | 8 (8–8) | 8 (8–8) | 8 (8–8) |
| | Min–Max | 0–8 | 0–8 | 0–8 | 0–8 | 0–8 |

**Table 1.** *Cont.*

| Baseline Characteristics | Variable Value | Negative OBSP Screening | Positive OBSP Screening | Negative Non–OBSP Screening | Positive Non–OBSP Screening | Total |
|---|---|---|---|---|---|---|
| | Death | 1218 (0.1%) | 493 (0.3%) | 220 (0.1%) | 135 (0.3%) | 2066 (0.1%) |
| | 75th birthday | 11,661 (1.0%) | 1340 (0.7%) | 2488 (1.5%) | 505 (1.2%) | 15,994 (1.0%) |
| Censoring reasons | Breast cancer diagnosis (except DCIS) | 11 (0.0%) | 7732 (4.0%) | 13 (0.0%) | 1854 (4.3%) | 9610 (0.6%) |
| | Breast implants | 6 (0.0%) | * 1–5 | 22 (0.0%) | * 33–37 | 66 (0.0%) |
| | Mastectomy | 15 (0.0%) | * 318–322 | 8 (0.0%) | * 89–93 | 434 (0.0%) |
| | End of episode | 1,126,298 (98.4%) | 184,644 (94.4%) | 159,413 (97.8%) | 40,337 (93.3%) | 1,510,692 (97.7%) |
| | End of OHIP eligibility | 501 (0.0%) | 112 (0.1%) | 114 (0.1%) | 26 (0.1%) | 753 (0.0%) |
| | LTC admission | 4732 (0.4%) | 1051 (0.5%) | 753 (0.5%) | 235 (0.5%) | 6771 (0.4%) |

SD = standard deviation, Q1 = first quartile, Q3 = third quartile, DCIS = ductal carcinoma in situ. * = Range of values provided due to small cell suppression and/or preventing back-calculations.

### 3.2. Costs

Figure 1 presents the average total costs per woman per month for the entire, OBSP and non-OBSP cohorts. The average total cost per woman per month was $136 ± $103, $134 ± $103 and $155 ± $104 for the entire, OBSP and non-OBSP cohorts, respectively. This was further disaggregated into the average total screening cost per month, which was $103 ± $8, $100 ± $4 and $117 ± $9 per woman for the entire, OBSP and non-OBSP cohorts, respectively, and the average total diagnostic cost per woman per month at $219 ± $166, $228 ± $165 and $178 ± $159.

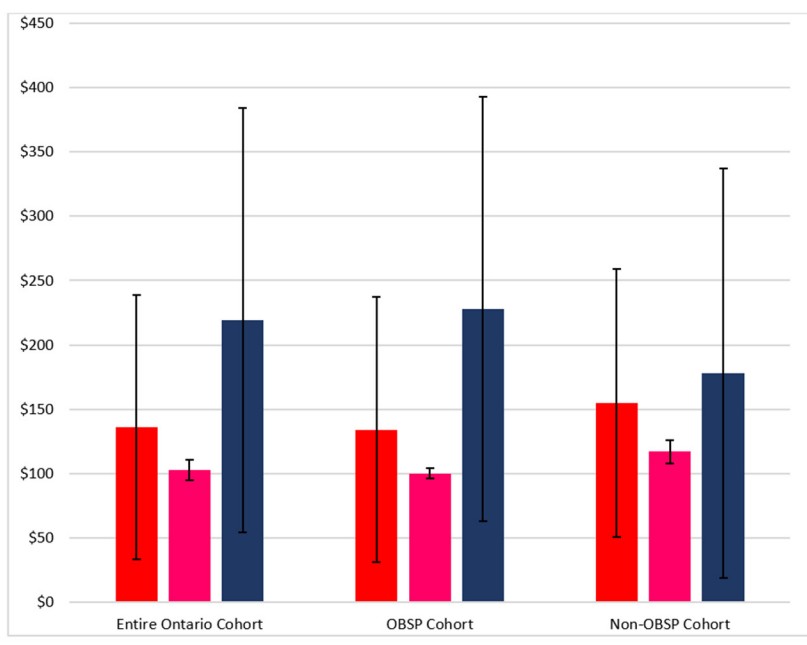

| Variable | Average Cost Per Woman Per Month ± Standard Deviation (N) | | |
|---|---|---|---|
| | **Entire Ontario Cohort** | **OBSP Cohort** | **Non-OBSP Cohort** |
| ■ Total Cost | $136 ± $103 (1,546,386) | $134 ± $103 (1,340,140) | $155 ± $104 (206,246) |
| ■ Total Screening Cost | $103 ± $8 (1,546,386) | $100 ± $4 (1,340,140) | $117 ± $9 (206,246) |
| ■ Total Diagnostic Cost | $219 ± $165 (238,621) | $228 ± $165 (195,439) | $178 ± $159 (43,182) |

**Figure 1.** Average Costs Per Month for Entire, OBSP and Non-OBSP Cohorts.

Figure 2A,B provide the breakdown of the screening costs and the diagnostic costs per month by their components for the entire Ontario cohort. The largest cost driver for screening was the actual cost of the mammogram procedure at $64 ± $6, followed by the

overhead cost at $51 ± $0. The largest cost drivers for the diagnostics were the biopsy cost at $225 ± $167 and the CT/MRI cost at $200 ± $65.

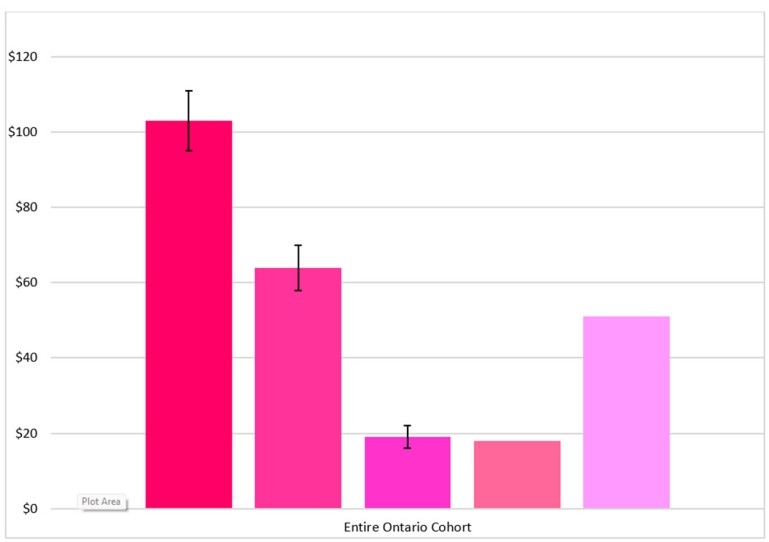

| Variable | Average Cost Per Woman Per Month ± Standard Deviation (N) |
|---|---|
| ■ Total screening cost | $103 ± $8 (1,546,386) |
| ■ OHIP screening cost | $64 ± $6 (1,546,386) |
| ■ Screening facility cost | $19 ± $3 (1,337,049) |
| ■ Screening administrative cost | $18 ± $0 (1,340,140) |
| ■ Overhead screening cost | $51 ± $0 (206,246) |

**(A)**

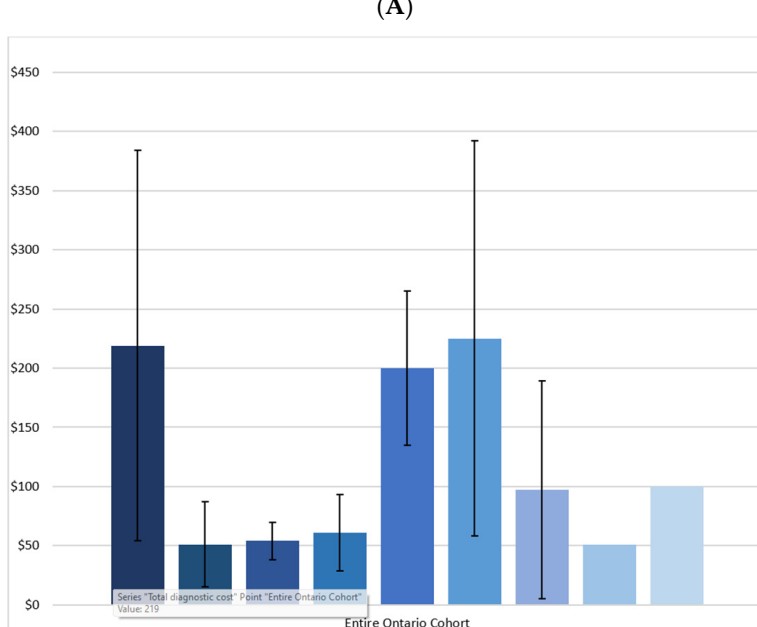

| Variable | Average Cost Per Woman Per Month ± Standard Deviation (N) |
|---|---|
| ■ Total diagnostic cost | $219 ± $165 (238,621) |
| ■ Standard diagnostic mammogram cost | $51 ± $36 (154,772) |
| ■ Specialized diagnostic mammogram cost | $54 ± $16 (4,923) |
| ■ Ultrasound cost | $61 ± $32 (184,503) |
| ■ CT/MRI cost | $200 ± $65 (5,579) |
| ■ Biopsy cost | $225 ± $167 (44,117) |
| ■ Diagnostic genetics cost | $97 ± $92 (2,151) |
| ■ Overhead diagnostic cost | $51 ± $0 (238,621) |
| ■ Diagnostic follow-up cost | $100 ± $0 (92,937) |

**(B)**

**Figure 2.** (**A**) Average Screening Costs Per Woman Per Month for the Entire Ontario Cohort. (**B**) Average Diagnostic Costs Per Woman Per Month for the Entire Ontario Cohort.

Table 2 presents the per women per month costs of the individual components of the screening and diagnostic costs stratified by the OBSP and non-OBSP cohorts. Similar to the entire 1.5 million cohort, the largest cost driver for screening was the actual cost of the screening mammogram at $64 ± $5 and $66 ± 9 for the OBSP and non-OBSP cohorts, respectively. The largest cost driver for diagnostics was the biopsy cost at $224 ± $165 and $233 ± $174 for the OBSP and non-OBSP cohort, respectively, and the CT/MRI cost at $198 ± $63 and $203 ± $67.

**Table 2.** Screening and Diagnostic Components for OBSP and Non-OBSP Cohort.

| | OBSP (Average Cost per Woman ± Standard Deviation (N)) | Non-OBSP (Average Cost per Woman ± Standard Deviation (N)) |
|---|---|---|
| Screening Costs | | |
| Total screening cost | $100 ± $4 (1,340,140) | $117 ± $9 (206,246) |
| Screening cost | $64 ± $5 (1,340,140) | $66 ± $9 (206,246) |
| Screening facility cost | $19 ± $3 (1,337,049) | N/A |
| Screening administrative cost | $18 ± $0 (1,340,140) | N/A |
| Overhead screening cost | N/A | $51 ± $0 (206,246) |
| Diagnostic Costs | | |
| Total diagnostic cost | $228 ± $165 (195,439) | $178 ± $159 (43,182) |
| Standard diagnostic mammogram cost | $51 ± $36 (137,010) | $50 ± $38 (17,762) |
| Specialized diagnostic mammogram cost | $54 ± $16 (4529) | $56 ± $19 (394) |
| Ultrasound cost | $60 ± $31 (148,525) | $67 ± $32 (35,978) |
| CT/MRI cost | $198 ± $63 (3368) | $203 ± $67 (2211) |
| Biopsy cost | $224 ± $165 (36,985) | $233 ± $173 (7132) |
| Diagnostic genetics cost | $100 ± $95 (1640) | $89 ± $80 (511) |
| Overhead diagnostic cost | $51 ± $0 (195,439) | $51 ± $0 (43,182) |
| Diagnostic follow-up cost | $100 ± $0 (92,937) | N/A |

*3.3. Sensitivity Analysis*

The screening and diagnostic unit costs in Quebec were lower and higher, respectively when compared to Ontario, resulting in a lower average total cost per Quebec woman per month ($123 ± $104) than those from Ontario ($136 ± $103; see Figure 3).

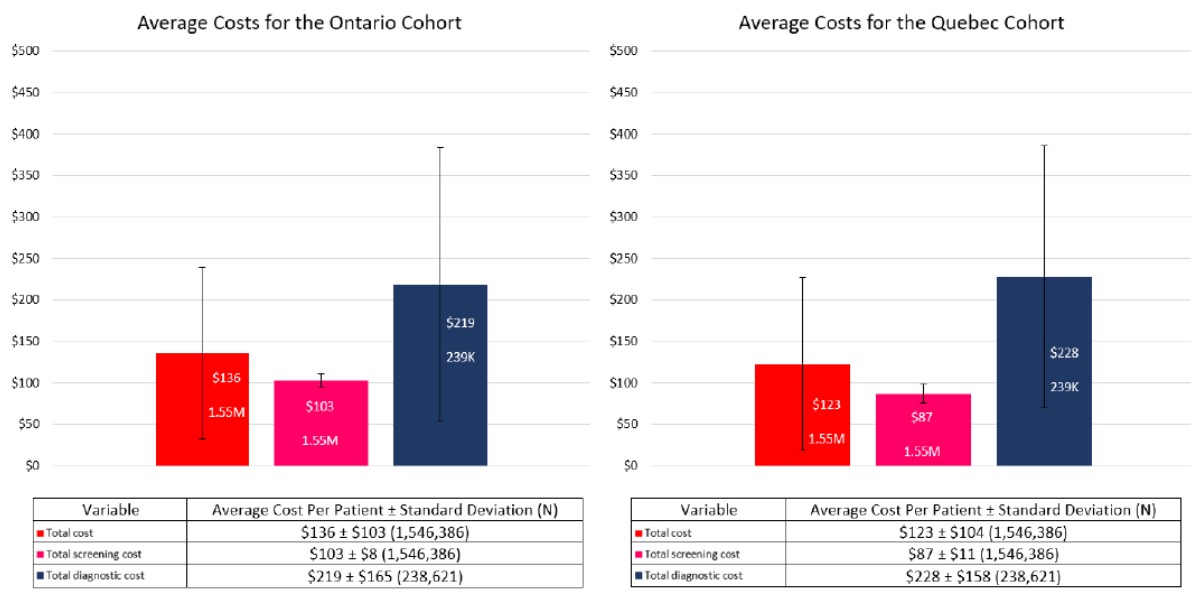

**Figure 3.** Comparison of Ontario and Quebec Average Cost Per Woman Per Month.

## 4. Discussion

Our results have shown that women in both the OBSP and non-OBSP cohorts, had similar distributions in terms of age and Charlson comorbidity scores. There were differences between the cohorts when examining costs. Women in the OBSP cohort have lower screening costs per month than women in the non-OBSP cohort ($100 versus $117, respectively; *p*-value < 0.001), however, women in OBSP incur much higher diagnostic costs per month than women in the non-OBSP cohort ($228 versus $178, respectively; *p*-value < 0.001). The reason why the OBSP cohort has lower screening costs is not due to the cost of the procedure itself being less expensive, rather it is the differences in the overhead and administrative costs that result in the non-OBSP cohort having a higher total screening cost. Similarly, women in the OBSP and non-OBSP cohorts have similar diagnostic procedure costs, except for genetics visits where women in the OBSP incur almost nearly $10 higher costs per month, however, women in the OBSP group also have a $100 diagnostic follow-up cost per month which is not present in the non-OBSP group. As a sensitivity analysis, unit costs from Quebec were used for both screening and diagnostic procedures that were lower and higher, respectively than those in Ontario; therefore, the lower average cost per Quebec woman was not surprising. However, in order to represent a more accurate representation of the Quebec breast cancer screening, actual total number of Quebec women being screened and those who receive follow-up diagnostic testing is needed and is planned in subsequent analyses.

Since our study included both the screening and diagnostic components of breast cancer screening as well as administrative costs, our results are more comprehensive in comparison cost-effectiveness studies of breast screening programs that often cite only the cost of the screening mammogram or approximate cost per woman of the entire screening program [4,5,10–13].

The strengths of this study include ensuring the use of data from both women screened within an organized program (OBSP) and those screened outside the OBSP (OHIP) have been captured. Secondly, in addition to costs associated with the screening and diagnostic procedures themselves, we have incorporated overhead costs, site-specific costs, and tomography costs. This study also has its limitations. We excluded women who had a breast cancer diagnosis that may have been a false positive. Although ultrasounds and MRI scans could have been used for breast cancer screening, this study limited its scope to only mammography screening to reflect OBSP funding policy for average risk women. Furthermore, while overhead and administrative costs were included in our analysis that are often excluded in other analyses, the overhead cost of $51 was a proxy based on Ontario Case Costing's indirect cost of breast imaging procedures performed institutionally. It is possible that the actual overhead cost could be lower if increased screening and diagnostic mammograms resulted in improved processes.

Future analyses would involve adapting the current costing algorithm for breast screening to reflect the different provincial screening programs and incorporating all relevant screening and diagnostic costs.

In conclusion, we determined monthly breast screening cost results stratified by OBSP versus non-OBSP and negative versus positive screens for average risk women being screened in Ontario, Canada with an 8-months follow-up period after the screening mammogram. Additional stratifications to be conducted will be by false negatives and false positives. Further analyses being considered are integrating risk stratifications and impact of genetic testing to screening costs.

**Supplementary Materials:** The following supporting information can be downloaded at: https://www.mdpi.com/article/10.3390/curroncol29110657/s1, Figure S1: Breast cancer screening schematic; Table S1: Exclusion codes; Table S2: Ontario specific diagnostic codes and unit costs; Table S3: Ontario specific cost assumptions; Table S4: Quebec specific unit costs.

**Author Contributions:** Conceptualization, N.M., S.J.S., C.D. and J.M.G.; Methodology, N.M., S.J.S., C.D., J.M.G., J.S. and A.M.C.; Software, C.D.; Validation, N.M., S.J.S., C.D., J.M.G., J.S. and A.M.C.; Formal Analysis, C.D.; Investigation, N.M., S.J.S., C.D., J.M.G., J.S. and A.M.C.; Resources, N.M., S.J.S., C.D., J.M.G., J.S. and A.M.C.; Data Curation, C.D.; Writing—Original Draft Preparation, S.J.S.; Writing—Review and Editing, N.M., S.J.S., C.D., J.M.G., M.W., M.-H.G., N.P., J.S. and A.M.C.; Visualization, S.J.S.; Supervision, S.J.S.; Project Administration, S.J.S.; Funding Acquisition, N.M., J.S. and A.M.C. All authors have read and agreed to the published version of the manuscript.

**Funding:** This study received funding from an unrestricted by the Ontario Research Fund via the University of Toronto. This work was funded by The PERSPECTIVE I&I project through the Government of Canada through Genome Canada (#13529) and the Canadian Institutes of Health Research (#155865), the Ministère de l'Économie et de l'Innovation du Québec through Genome Québec, the Quebec Breast Cancer Foundation, the CHU de Quebec Foundation and the Ontario Ministry of Research and Innovation through the Ontario Research Fund. This study was supported by ICES, which is funded by an annual grant from the Ontario Ministry of Health (MOH) and the Ministry of Long-Term Care (MLTC). This study also received funding from the Canadian Institutes of Health Research (CIHR). The analyses, conclusions, opinions and statements expressed herein are solely those of the authors and do not reflect those of the funding or data sources; no endorsement is intended or should be inferred.

**Institutional Review Board Statement:** For this study, #1849 is the project identification number given by the Sunnybrook Research Institute's Research Ethics Board last dated 14 May 2022 and will expire on 10 June 2023.

**Informed Consent Statement:** Not applicable.

**Data Availability Statement:** Parts of this material are based on data and information compiled and provided by the Canadian Institute for Health Information (CIHI), the Ontario Ministry of Health (MOH), Ontario Health (OH), and Statistics Canada. The dataset from this study is held securely in coded form at ICES. While legal data sharing agreements between ICES and data providers (e.g., healthcare organizations and government) prohibit ICES from making the dataset publicly available, access may be granted to those who meet pre-specified criteria for confidential access, available at https://www.ices.on.ca/DAS (accessed 20 October 2022; email: das@ices.on.ca). The full dataset creation plan and underlying analytic code are available from the authors upon request, understanding that the computer programs may rely upon coding templates or macros that are unique to ICES and are therefore either inaccessible or may require modification.

**Conflicts of Interest:** The authors declare no conflict of interest.

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
