# Peer review of "Using Real-World Data to Determine Health System Costs of Ontario Women Screened for Breast Cancer"

_curroncol, doi:10.3390/curroncol29110657_

Round 1
Reviewer 1 Report
Dear Authors,
thanks for your research paper. The paper is very clear structured, focussed and based on a huge empirical data base.
I have some minor remarks:
In the abstract line 16 you write "49 and 74 years". "and" is wrong.
Cost effectiveness means to examine both the costs and health outcomes of one or more interventions. It compares an intervention to another intervention (or the status quo) by estimating how much it costs to gain a unit of a health outcome. For me it is not clear, what kind of health outcome you underline. Screening results can be not correct and therefore a screening program could be low cost but higher incorrect results. I supervise to discuss the two programs (OBSP and Non-OBSP) a little bit about their outcomes.
At the end of line 58 there is missing the point at the end of the sentence.
There are problems using the overhead cost and the cost of administration reported by the OBSP. For example usually you have an economies of scale effect: The more screenings the lower are the fixed cost like facility or administrative cost. And: With more screenings you are able to establish better processes (like digitalization and automation procedures and techniques) which leads to lower fixed costs.
The same problems you may have for the NON-OBSP overhead cost. I give you the advise to discuss this kind of limitations.
I have fun reading your paper. I wish you all the best for your career.
Reviewer 2 Report
Reference Report
This manuscript investigated the breast cancer screening costs in Ontario, Canada using the real-world data. The authors compared the costs between a formal (OBSP) and an informal (non-OBSP) screening program using an administrative database. They also compared their results with those in Quebec, another province in Canada. From the results, they found similar screening costs across different cohorts, but higher diagnostic costs for the OBSP cohort. This work is comprehensive and topical. However, I have the following concerns:
1. Abstract, L15: Please provide long form of “OBSP”.
2. Introduction, L32: The age range of “50-74” does not match with that mentioned in the Abstract: “49-74”.
3. Introduction, L36 and L38: “ICMS” or “ICES”? Also, please provide a reference for that CCO developed system.
4. Section 2.1: Please justify the selection of study period from Jan 1, 2013 to Dec 31, 2019 as the ICMS was developed in 2000.
5. Section 2.1 and 2.1: Both sections have the same title.
6. Section 2.2, L70-71: Please provide references or more information for the X172 and X178 quotes.
7. Section 2.2: I suggest using a simple flowchart to help the readers to understand the screening process through OBSP and non-OBSP.
8. Section 2.3, L115: What is “ICES”? please provide the long form. Is it the “ICES” in the Introduction section or it is a typo of “ICMS”?
9. Section 2.4, L136: Please provide the information of software to perform the statistical test (e.g. t-test).
10. Section 3.1, 152-153: Negative Non-OBSP is about 3 times of the Positive Non-OBSP. This is different from the negative OBSP which is about 5.7 times than the positive OBSP. What is the reason?
11. Section 3.2: Can you explain why the calculated SDs reduced a lot when considering the average total screening cost per month from the average total cost per woman per month?
12. Figure 1: It is good to make the columns transparent so that error bars inside the columns can be seen.
13. Figure 2A and 2B: Please explain why error bars are missing in the last two columns.
Round 2
Reviewer 2 Report
I am satisfied with the modifications/corrections from the authors as per my comments. I also accepted the explanation of the authors for my concerns in this work. The quality and presentation are improved in this revision.